# Development of a novel therapy for systolic heart failure

Corey Pollock[1], Xilun Wang[1], Hussam Alsaraji[1], Joseph Menassa[1], George Mbogo[1], Dimuthu Angage [1], Benjamin Richards[1], Jason Glab[1], Keshava K Datta [2], Liana Theodoridis [3], Steve Petrovski [3], Daniel Donner [4], Yuvixza Lizarme-Salas[1], Xiao-Jun Du [4], Michael Foley[1], Brian J Smith[1], Belinda Abbott[1 ✉] & Hamsa Puthalakath [1 ✉]

## Abstract

Heart failure presents a critical health challenge with a 5-year mortality rate of up to 50%. Conventional treatments often lead to bradycardia or hypotension due to their impact on patient hemodynamics. To address this issue, we utilized high-throughput drug screening combined with structure-activity relationship-based medicinal chemistry to develop a novel drug-like compound that effectively blocks the β-adrenergic receptor (β-AR) mediated apoptosis pathway. This compound demonstrated both safety and efficacy in pre-clinical mouse models without adversely affecting cardiac output. Through thermal proteome profiling mass spectrometry, we identified the compound's target as Wdr3, a regulator of the Hippo signaling pathway. This target identification was further validated using CRISPR-based knockout experiments. Our findings provide a valuable framework for the development of hemodynamically neutral therapies aimed at treating systolic heart failure.

**Keywords** Apoptosis; Bim; β-AR; HF; PKA
**Subject Categories** Cardiovascular System; Pharmacology & Drug Discovery

## Introduction

Heart failure (HF) is a condition where the heart is unable to pump blood to adequately perfuse vital organs to supply nutrients and oxygen. It can be the result of multiple myocardial diseases leading to impaired cardiac output (McDonagh et al, 2022). It is prevalent in ~1–2% of the adult population and up to 80% of the population above the age of 80. After the diagnosis, it has a 5-year mortality rate of 50%, irrespective of the aetiology or the treatment modality (Gerber et al, 2015). Conventionally, HF is classified based on the left-ventricular ejection fraction (LVEF): HF with reduced ejection fraction (HFrEF or systolic heart failure or SHF) and HF with preserved ejection fraction (HFpEF or diastolic heart failure or DHF) (Wybraniec et al, 2022). These two forms of HF are equally prevalent in the population. One of the main causes of SHF is dilated cardiomyopathy (DCM), this being the leading cause of heart transplantation globally (Tayal et al, 2017). Apoptotic cell death plays a major role in the dysregulation of cardiomyocyte homoeostasis leading to DCM (Hong et al, 2000); this process could be triggered by catecholamine imbalance as seen in takotsubo syndrome (Gupta et al, 2018) or in response to autoantibodies against β-adrenergic receptors (β-AR) as seen in Grave's disease (Jane-wit et al, 2007; Xia and Kellems, 2011).

Historically, the development of drugs to treat SHF has targeted the secondary consequences of the disease, including peripheral maladaptive processes and cardiac remodelling. Over the course of the last 25 years, a variety of drugs have been developed, such as drugs that target the renin-angiotensin system, the adrenergic system, neuro-hormonal blockers, channel blockers and various types of vasodilators (Lewis et al, 2017). To date, all these disease-modifying therapies cause hypotension and bradycardia as unintended consequences. Low systolic pressure due to reduced cardiac output or due to drug-induced vasodilation is one of the strongest determinants of the clinical course in patients with SHF (MacFadyen et al, 2003; Mehra et al, 2003). By leading to inadequate coronary, cerebral and renal perfusion, especially in high-risk patients, hypotension in HFrEF, presents a major hurdle in contemporary drug development, especially with the continued development of hemodynamically active drugs (Hsiao and Chu, 2018; Vishram-Nielsen et al, 2022). In past clinical trials, this drug side-effect was underestimated because the tools used were relatively insensitive to therapy-related hemodynamic disturbances and high-risk hypotensive patients were invariably excluded from such trials (Vaduganathan et al, 2015). Therefore, the development of new drugs for treating HFrEF must take into consideration variations in baseline blood pressure of the target population.

The protein Bim is an essential initiator of apoptosis in a wide variety of physiological settings (Doerflinger et al, 2015; Moujalled

[1]Department of Biochemistry and Chemistry, La Trobe University, Kingsbury Drive, Bundoora, Vic 3086, Australia. [2]Research Platforms, La Trobe University, Kingsbury Drive, Bundoora, Vic 3086, Australia. [3]Department of Microbiology, Anatomy and Physiology, La Trobe University, Kingsbury Drive, Bundoora, Vic 3086, Australia. [4]Baker Heart and Diabetes Institute, 75 Commercial Road, Victoria 3004, Australia. ✉E-mail: b.abbott@latrobe.edu.au; h.puthalakath@latrobe.edu.au

et al, 2011). We have previously reported that activation of β-ARs in cardiomyocytes leads to Bim-dependent apoptosis and HF. We have also elucidated the signal transduction pathway leading to Bim induction during this process (Lee et al, 2013). This knowledge has been used to generate novel methods of screening drug-like compound libraries and establishing activity validation strategies. Screening of a boutique compound library yielded several very promising hits that specifically targeted Bim at the mRNA level without affecting protein kinase A (PKA) activity, the protein needed for the contractile function of cardiomyocytes. Based on structure-activity relationship (SAR) analyses, we developed analogues and then tested them in vitro. We chose the drug lead with the most favourable profile (BR43) for further investigation. Through thermal proteome profiling mass spectrometry (TPP-MS), we identified BR43's target and confirmed this through gene knockout. BR43 has been tested in vivo for safety and for efficacy in treating mice with acute cardiomyopathy using the Takotsubo model. BR43 was found to be effective in controlling heart failure without compromising cardiac output, i.e. the drug-like compound is hemodynamically neutral. Furthermore, this compound also has been found to be as effective as beta-blockers in reducing anthracycline-induced cardiotoxicity.

# Results

## Screening strategy

The assay was set up with the rat embryonic cardiomyocyte cell line (H9C2) treated with dobutamine (β-AR agonist), rolipram (PDE inhibitor) and the test compounds. Viability was measured in the presence of the metabolic indicator dye resazurin, which is reduced to resorufin by viable cells and measured fluorometrically (579Ex/584Em) (Fig. 1A). We conducted a screen of ~5000 diverse, lead-like library compounds in 384-well microtiter plate format. Z'(Zhang et al, 1999) exceeded 0.4 for all 14 assay plates (Fig. EV1), indicating the cell-based assay's robustness and suitability for HTS. Figure 1B shows a histogram of the ~5000 compounds (y-axis) grouped into specific percent viability bins (x-axis). As expected, the average compound library result was close to 0, where 0 is the average result of the rolipram/dobutamine-treated wells, and 100 is the average result of the rolipram-only-treated wells. Applying a hit selection threshold of 2x standard deviation above the average, 35 compounds (highlighted in red in Fig. 1B) were selected for progression to validation steps. As the initial screen was a non-biased phenotypic screen, which could target any component of the β-AR pathway. Therefore, the drug hits identified in the initial screen were validated for specificity as follows:

## Validation through cell death analysis

The initial hits (35 compounds) were compared with Pindolol (a clinically used beta-blocker) for their effectiveness in blocking dobutamine (10 μM) mediated apoptosis in H9C2 cells using Annexin V staining (CellTiter-Blue assay used in the screen could potentially select auto-fluorescent compounds). Of the 35 compounds, 21 appeared to have a significant effect on survival comparable to that of Pindolol and were selected for further analysis (indicated by the dotted line in Fig. 1C).

## Validation through expression analysis

Stable clones of mouse embryonic fibroblasts (MEFs) expressing PKACα from a 4-OHT (4-hydroxytamoxifen)-inducible lentiviral vector robustly upregulate Bim (Fig. 1D). Since PKA activation in these cells is independent of β-AR, we reasoned that the compounds that block PKA-mediated Bim induction were not acting at the level of β-AR activation. These cells were treated with 10 μM of the 21 compounds to test their effect on Bim protein levels. Furthermore, assessing the phosphorylation status of one of the key substrates of PKA, i.e. CREB (phosphor-CREB) would determine if the compounds had any effect on PKA activation. As evident from Fig. 1E, five compounds (indicated by asterisks) passed this test, i.e. they inhibited PKA-mediated Bim induction without affecting PKA activity. The effect on Bim induction appeared to be specific to the PKA pathway, as Bim induction was unaffected by these compounds in response to thapsigargin-induced endoplasmic reticulum (ER) stress (Fig. 1F). Furthermore, we tested these compounds on their effect on β-AR activation-mediated Bim induction and found to have the same effect in H9C2 cardiomyocyte cell lines (Fig. 1G). These five compounds were tested further for their ability to inhibit Bim at the RNA level by absolute quantitation using droplet digital PCR. Except for C5 and C23, three compounds (C24, C25 and C26) caused a robust reduction in Bim mRNA levels (Fig. 1H). The two compounds (C5 and C23) that did not reduce Bim mRNA and C24 that required the use of potassium cyanide or derivatives with a similar toxicity profile were not further investigated. The synthesis and preliminary evaluation of C25 analogues has been reported previously (Richards et al, 2022), but they were found to have reduced potency in comparison with C26. As a result, only lead compound C26 was selected for further characterisation (Fig. 1I).

## Mode of action

We had previously proposed that the transcriptional induction of Bim during β-AR signalling involved epigenetic regulation through histone acetylation at the Bim promoter (Lee et al, 2013). However, we found no evidence for these compounds inhibiting acetylation using anacardic acid as the positive control for acetylation inhibition (Fig. EV2). Furthermore, c-Myc transcriptional activity could be stimulated by acetylation through the recruitment of the cofactor CBP (Vervoorts et al, 2003). We tested this possibility in 293T transient expression and immunoprecipitation assays. While CBP could clearly acetylate c-Myc, this was not impacted by the presence of these compounds (Fig. EV3). Therefore, we conclude that the transcriptional inhibition of Bim by these drug-like compounds is not mediated through inhibition of acetylation.

## SAR and the development of BR43

To explore the structure-activity relationships of C26 (3-(pyrrolidin-2-yl) isoxazole), a series of analogues was designed and synthesised. The synthesis of C26 was undertaken using a convergent strategy of two starting materials independently prepared (EV4). The required bromomethylindole 1 was generated for the N-alkylation of the 3-(pyrrolidine-2-yl) isoxazole 2. Compound 3 was then subjected to Boc-deprotection to generate compound 4 and a second N-alkylation using

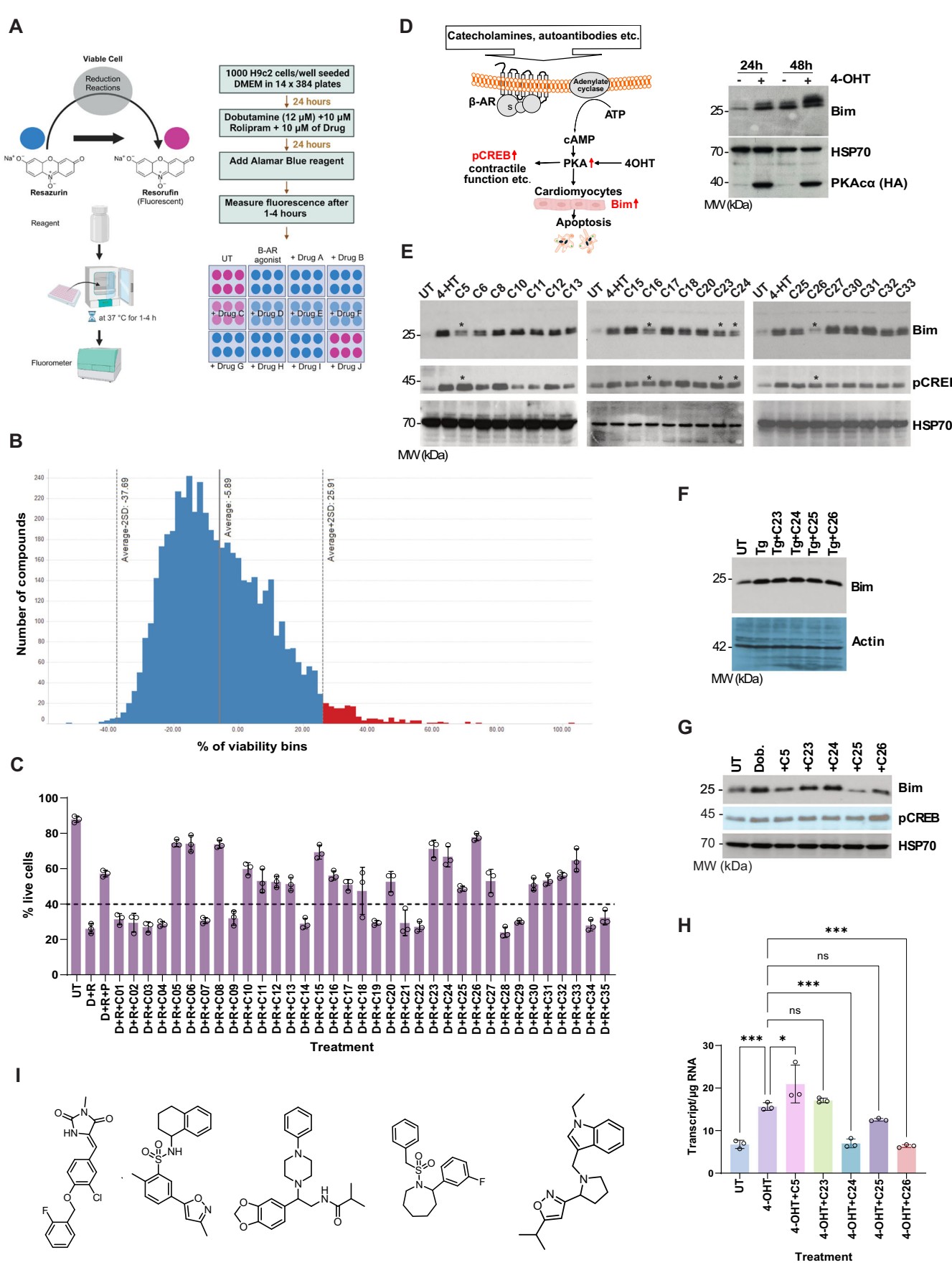

**Figure 1. Identification of C26 as the drug scaffold.**

(A) A schematic of the drug screening strategy to identify drugs that block β-AR-mediated cell death in rat H9c2 cells. (B) The high-throughput screening results. Histogram showing number of compounds on the y-axis and percent viability bins on the x-axis. 35 compounds above the selection threshold, i.e. 2x standard deviation above the average, were chosen for further validation. (C) Validation through apoptosis assay using the beta-blocker pindolol (10 μM) as the control (lane D + R + P, i.e. dobutamine, rolipram, and pindolol) in H9c2 cells. Error bars are ±SEM, *N* = 3 independent assays. (D) The assay system for drug validation for specific Bim inhibition (The schematic on the left). MEFs expressing the catalytic subunit of PKA (PKACα) under 4-OHT regulation specifically induce Bim independent of β-AR activation, as determined by Western blot. (E) The 21 compounds that passed apoptosis assay validation were further tested for their ability to block Bim without impacting the surrogate marker for PKA activity, i.e. pCREB levels. The cells were treated with 100 nm of 4-OHT in the presence or in the absence of 10 μM of each of the test compounds for 24 h before Western blot analysis. Cells with no 4-OHT and the drugs were used as a control (UT). (F) Testing of the compounds selected (shown in asterisks in Fig. 1E) for their specificity for the β-AR pathway, MEFs were treated with thapsigargin to induce ER stress in the presence or in the absence of the four compounds as indicated, and Bim levels were measured by Western blot. (G) Validation of the compounds in H9C2 cells treated with the β-AR agonist dobutamine (Dob) in the presence of the five compounds by Western blot. (H) The five compounds were further tested for their ability to block Bim induction at mRNA levels by digital droplet PCR. Error bars ± SEM, *N* = 3 independent experiments, *\*p* = 0.0131-0.0378, \*\*\*p = 0.0001–0.0009, One-way ANOVA with Tukey's multiple comparisons. (I) The chemical structures of the five compounds. Source data are available online for this figure.

ethyl bromide to complete the synthesis of C26. A total of 16 analogues were produced using this strategy by varying the two starting materials or the alkylating agent used for the final step (See the appendix information for the details on syntheses).

When biological evaluation of the analogue series was undertaken using Western blot assays, several compounds induced a downregulation of Bim similar to that demonstrated by C26 (Fig. 2A). These structure-activity relationships are summarised in Fig. 2B. Only the piperidine analogue BR43 (Fig. 2B), showed increased potency compared to C26, the initial lead compound. It appears that expansion from a five to a six-membered ring was responsible for this improved potency. There may be further scope for improving the potency by synthesising analogues that preserve the piperidine moiety.

## In vivo safety assessment of BR43

Of the various C26 analogues that could downregulate PKA-mediated Bim induction, BR43 had minimal effect (even at 24 μM) on cellular proliferation as measured by the Sulforhodamine assay (Fig. 2C). Further in vitro analysis using H9C2 cells found BR43 to be as effective as Pindolol (a clinically approved beta-blocker) in inhibiting β-AR-mediated Bim induction (Fig. 2D), and, therefore, BR43 chosen for in vivo safety analyses. C57B/6 mice (six mice in each group) were treated (IP) with 0, 0.26, 0.53 and 0.79 mg/kg body weight (which was equivalent to 0, 10, 20 and 30 μM of the drug, assuming an average blood volume of 1.5 ml) daily for seven days. Observations of body conditions and weight measurements were taken daily. At the end of the experiment, mice were euthanised, and blood and tissues were harvested for histological and haematological analyses. No significant differences were found between treatments (Fig. EV5), and therefore we concluded BR43 to be safe for further experiments.

## In vivo assessment in a heart failure model

To assess the efficacy of the drug, we used the clinically relevant HF model, the takotsubo cardiomyopathy model. This is a widely used model that mimics myocardial infarction and HF (Lee et al, 2013; Shao et al, 2013). Isoproterenol (1.25 mg/kg/h) was administered to C57B/6 mice by Alzet mini osmotic pump for 24 h (Sham animals had PBS delivered by the osmotic pump) and the animals were given either vehicle (DMSO) or BR43 (0.30 mg/kg) via intraperitoneal injection for the duration of the experiment i.e. 7 days post-

catecholamine shock. Subsequently, the heart rate and ejection fraction in these mice were then assessed by echocardiography (Fig. 3A,B). While there was no significant difference in the heart rate and end diastolic volume between the three groups (i.e. untreated, isoproterenol treated and isoproterenol + BR43 treated), there was a significant difference in the end systolic volume and in the left-ventricular ejection fraction (LVEF) observed between the three groups, demonstrating the effectiveness of the drug in protecting from catecholamine-induced systolic HF (Fig. 3A). Moreover, tissue analysis by Western blot revealed that BR43 reduced Bim induction levels mediated by isoproterenol treatment while maintaining pCREB levels. This confirmed that BR43 could protect mice from cardiomyopathy without compromising the contractile function and, therefore, the cardiac output (Fig. 3C).

## In vivo assessment in a cardiotoxicity model

Anthracyclines are a class of drugs used in treating various types of cancers, however with various side effects, particularly cardiotoxicity leading to systolic heart failure. Beta-blockers are commonly used to mitigate the cardiotoxicity (He et al, 2022). Therefore, we tested if BR43 could be used for treating doxorubicin-induced cardiotoxicity (Herman et al, 1998). A cumulative dose of 15 mg/Kg over a period of 12 days in C57B/6 mice (Liu et al, 2012) resulted in a robust induction of heart failure as indicated by the troponin levels in the serum (Fig. 3D). This was reduced to below detection levels with a simultaneous administration of BR43 or atenolol (0.3 and 0.5 mg/Kg, respectively) on alternate days for 12 days (Fig. 3D).

## Identification of the drug target

To identify the target of BR43 in the cell, we employed thermal proteome profiling (TPP) mass spectrometry (Franken et al, 2015). TPP mass spectrometry indicated that Wdr3 (WD repeat-containing protein 3) is a potential target (Fig. 4A), where a temperature-dependent, accelerated denaturation was observed for Wdr3 in the presence of BR43. This was verified through Western blot analysis of HEK293T cells (Fig. 4B). Wdr3 has been reported to be involved in a variety of cellular processes such as genome stability, proliferation, signal transduction and apoptosis (Su et al, 2021). Importantly, Wdr3 is involved in the Hippo signalling pathway (Su et al, 2021), and we previously reported that the Hippo pathway is a regulator of Bim expression in a chronic heart failure model (Zhao et al, 2019). Therefore, we generated CRISPR

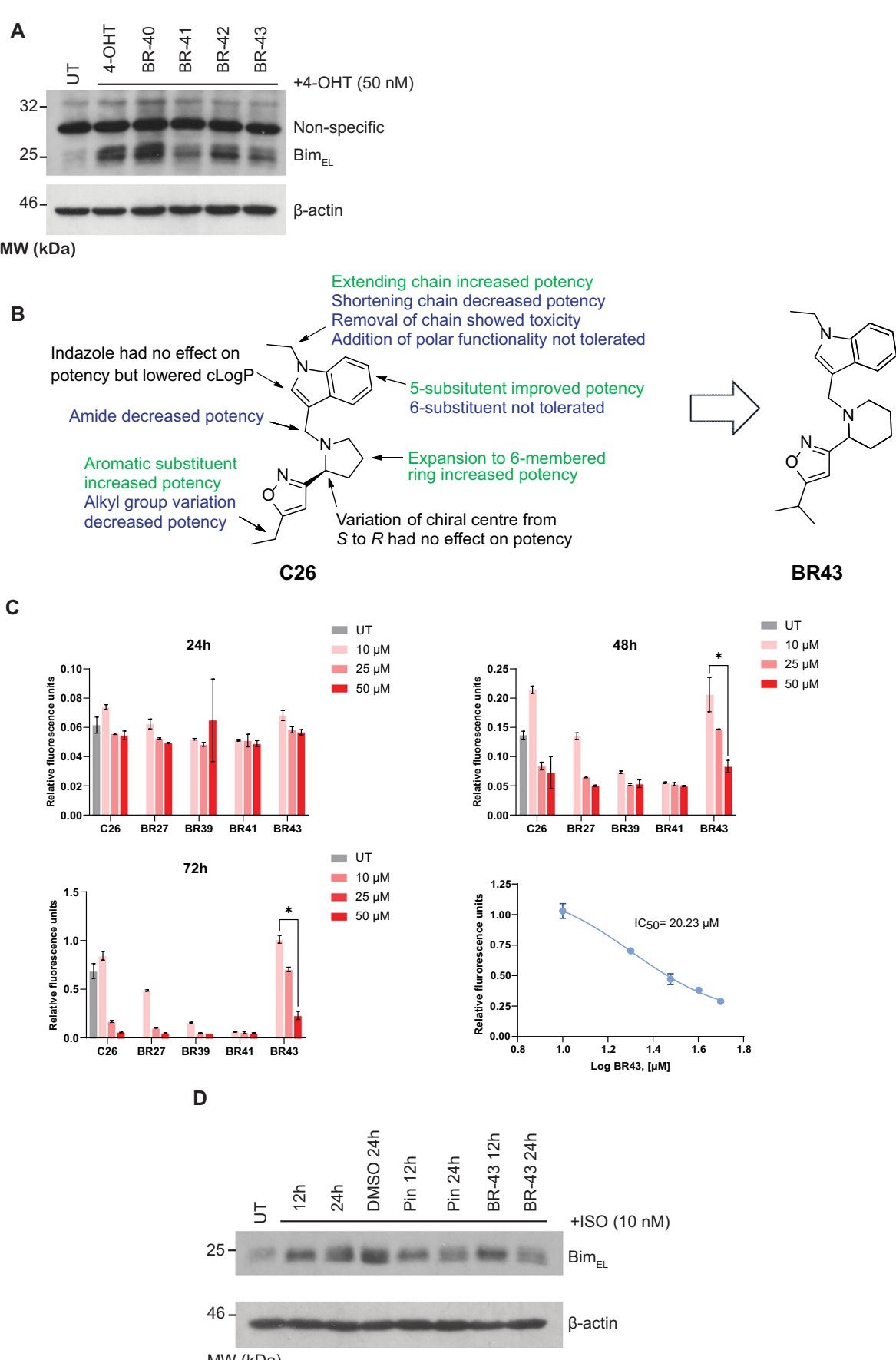

**Figure 2. SAR analysis of the C26 scaffold.**

(A) Western blot analysis of some of the derivatives of the C26 scaffold demonstrating their effectiveness in reducing Bim levels during PKA activation. The cells were treated with 100 nM of 4-OHT in the presence or in the absence of 10 μM of each of the test compounds for 24 h before Western blot analysis. Cells with no 4-OHT and the drugs were used as a control (UT). (B) The chemical structure of BR43 and a summary of various modifications on the C26 scaffold and their effect on the potency. (C) Effect of various C26 derivatives on cell proliferation as determined by sulforhodamine assay. The $IC_{50}$ was determined after 72 h of incubation. Error bars ± SEM, $N = 3$ independent experiments, $*p = 0.0027$, two-way ANOVA with Tukey's multiple comparison test. (D) Effect of BR43 on Bim levels in H9C2 cells under β-AR stimulation with Pindolol (10 μM) as the control. Source data are available online for this figure.

knock-out cell lines in our experimental system i.e. MEFs expressing inducible PKA catalytic subunit (PKACα). Since there were no antibodies that react with murine Wdr3 available, we sequenced the *Wdr3* locus, which was targeted to confirm the gene deletion (Fig. 4C). Consistent with the TPP results, genomic deletion of *Wdr3* resulted in the down regulation of Bim expression under PKA activation (Fig. 4D). However, this was restricted only to the PKA pathway as thapsigargin-mediated ER stress resulted in comparable Bim induction in all these cells (Fig. 4E).

## Molecular modelling

Further insights into the BR43-Wdr3 interaction were obtained by performing molecular docking simulations to predict the binding modes and affinities. Wdr3 is a component of the small subunit (SSU) processome, the structure of which was recently determined by cryo-EM. BR43 was docked against the Wdr3 domain of the human small-subunit processome with a weak predicted binding affinity of 34 μM. There are no known small molecule ligands that bind Wdr3; however, the SAM (S-adenosylmethionine) and SAH (S-adenosylhomocysteine) metabolites bind the ribosomal RNA small-subunit methyltransferase NEP1 of the SSU. SAM and SAH share similar molecular topology with BR43, a central cationic moiety (tertiary ammonium/sulfonium), and a 5:6 ring-fused heterocyclic extension (indole/adenine). Docking of SAM and SAH to Wdr3 and subsequent molecular dynamics (MD) yielded predicted binding affinities of 4 and 5 μM, respectively. The structures of SAM and BR43 were docked to NEP1 by molecular replacement of SAH. SAM and SAH were predicted to bind NEP1 with high affinity, 0.9 and 0.3 μM, respectively, whereas BR43 was predicted to bind more tightly, 0.1 μM (Figs. 5A and EV6).

Finally, we tested the impact of SAM and its derivative S-adenosylhomocysteine (SAH) on Bim induction during PKA signalling. Both SAM and SAH induced Bim synergistically during PKA signalling (Fig. 5B) and this induction was dependent on Wdr3 as Bim induction was significantly reduced in Wdr3-deficient cells (Fig. 5C), further confirming the role of Wdr3 in PKA-mediated Bim induction.

## Discussion

Though recent advances for treating systolic heart failure have focused on novel approaches to improve patient outcomes, they still rely on modulating the hemodynamic process. One notable advance is the emergence of sacubitril/valsartan, a first-in-class angiotensin receptor-neprilysin inhibitor (ARNI), which has demonstrated superior efficacy compared to traditional renin-angiotensin-aldosterone system (RAAS) inhibitors in large-scale

clinical trials. ARNI therapy combines the benefits of angiotensin receptor blockade with neprilysin inhibition, resulting in vasodilation, natriuretic effects, and reduced cardiac remodelling. However, hospital admission due to hypotension following sacubitril/valsartan administration has been reported (Hsiao and Chu, 2018; Vardeny et al, 2018). Similarly, sodium-glucose cotransporter 2 (SGLT2) inhibitors, initially developed for diabetes management, have shown remarkable cardiovascular benefits in heart failure patients but have had the same hemodynamic consequences (Rong et al, 2020). Furthermore, despite these recent advances, the 5-year mortality rate in systolic HF (HFrEF) has been reported to be as high as 75% (Shahim et al, 2023). Similarly, omecamtiv mecarbil (OM), a myosin activator with inotropic effects without altering calcium homoeostasis showed great promise (Zhou et al, 2024) but was denied FDA approval due to lack of efficacy in clinical trials. In this context, our attempt to develop a hemodynamically neutral therapy for HFrEF has added significance and may represent a pivotal advance in the development of cardiovascular therapeutics. We believe that the template we provided will pave the way for the development of a new class of drugs with high potency and selectivity that will modulate the underlying pathophysiological processes of HF without significantly affecting the hemodynamics. This approach could also be useful in developing novel drugs for treating chemotherapy-induced cardiotoxicity.

Our approach of screening for drug-like compounds that would selectively inhibit the pro-apoptotic protein Bim was built on our previous work, which demonstrated that Bim-deficient mice were resistant to catecholamine-induced cardiomyopathy (Lee et al, 2013), despite the onset of physiological hypertrophy and increased β-AR expression in the heart (Glab et al, 2017). Though, the initial screening was based on classical β-AR stimulation and cell survival, the validation of the drug hits was facilitated greatly using MEFs expressing the catalytic subunit of PKAα (PKACα) under 4-OHT induction. This helped us to validate the hits, solely based on their ability to downregulate Bim (that is responsible for cardiomyocyte attrition during βAR-signalling) without any impact on β-AR signalling and PKA activation (Fig. 6). Medicinal chemistry and SAR analyses helped us to derive BR43, which appears to have little cytostatic or cytotoxic effect but at the same time was able to reduce Bim induction and cardiomyopathy in a clinically relevant mouse model. Identification of the drug target through TPP, which was confirmed by using genetic knockouts, is consistent with our previous findings on the role of the Hippo pathway in inducing Bim and leading to cardiomyopathy (Zhao et al, 2019). However, the precise role of Wdr3 on Bim regulation remains to be elucidated, and further modifications of the drug scaffold is needed to improve the efficacy. The present work will provide a template for further work to develop hemodynamically neutral drugs for treating HFrEF.

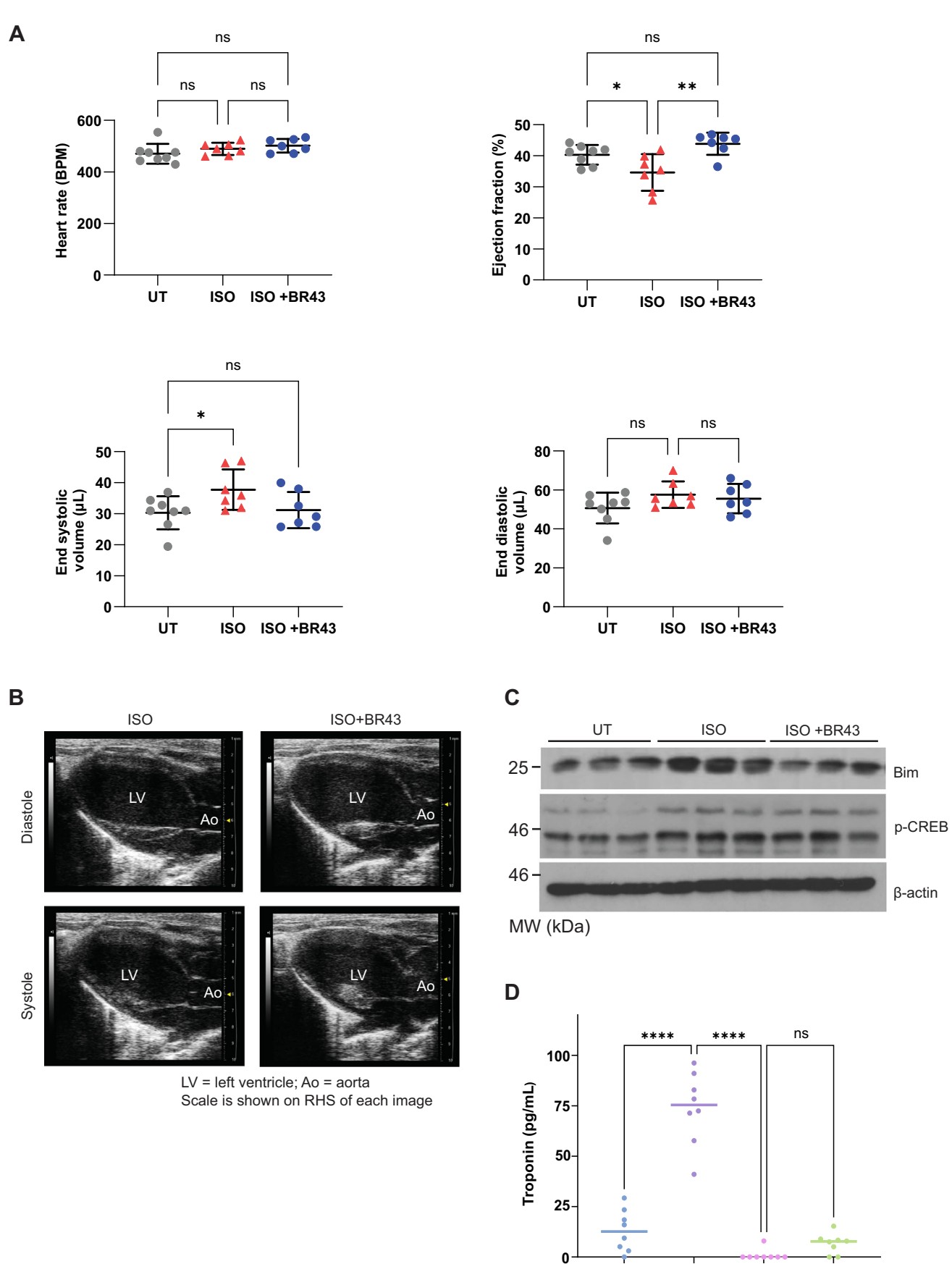

LV = left ventricle; Ao = aorta
Scale is shown on RHS of each image

**Figure 3. Analysis of BR43 efficacy in vivo cardiomyopathy models.**

(A) Mice were administered a single dose of isoproterenol (1.25 mg/kg/h) for 24 h, followed by BR43 (0.30 mg/kg) for 7 days or vehicle control. Mice were assessed for heart rate, end systolic/diastolic volumes and ejection fraction after day 1. Error bars ± SD, $n = 7-8$ animals in each group, *$p = 0.0419-0.0491$, ***$p = 0.0021$, one-way ANOVA with Tukey's multiple comparison test. (B) Representative echocardiogram images of mice either treated with isoproterenol plus vehicle control or isoproterenol plus BR43. (C) On day 8, mice were euthanised and heart tissues were subjected to Western blot analysis for Bim and pCREB. (D) Mice were administered with doxorubicin or dox plus BR43 or Dox plus atenolol, and serum samples were analysed for troponin levels by ELISA. Error bars ± SD, $n = 8$ animals in each group, ***$p < 0.0001$, one-way ANOVA with Tukey's multiple comparison test. Source data are available online for this figure.

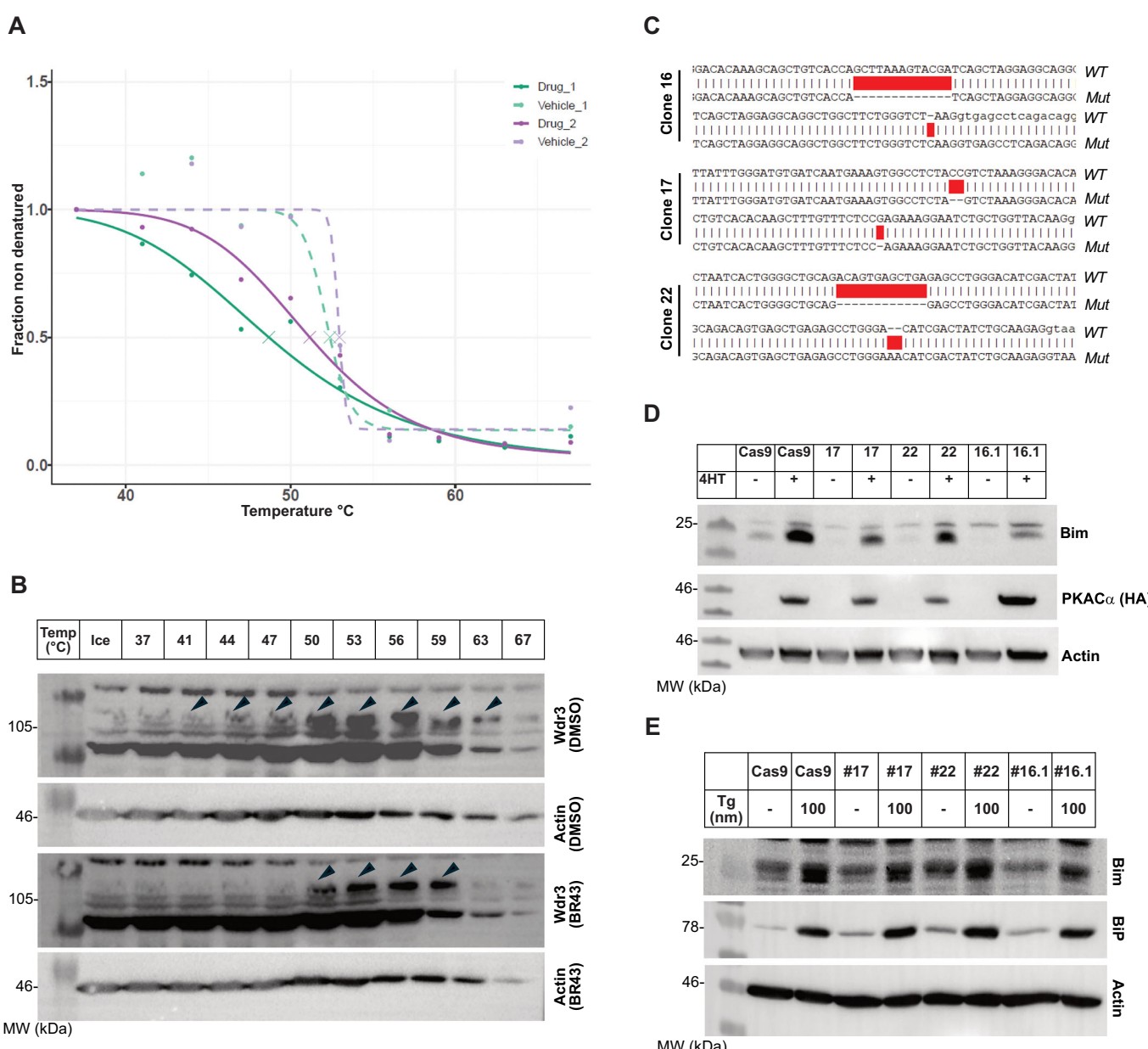

**Figure 4. Drug target identification by Thermal Proteome Profiling assay (TPP).**

(A) The TPP mass spectrometry profile of Wdr3 in the presence of BR43. Enhanced solubilisation of Wdr3 is seen in the presence of BR43 compared to vehicle control DMSO (From 37 to 50 °C). (B) Western blot analysis of 293T cells subjected to thermal profiling by Western blot. Arrowheads indicate the Wdr3-specific band undergoing increased solubilisation in the presence of BR43 compared to DMSO. (C) Sequence alignment of three independent CRISPR knockout clones of Wdr3. (D) Wdr3 knockout clones have significantly reduced Bim induction compared to the parental Cas9 clone, during PKA activation by 4-OHT treatment (100 nM) for 24 h. Anti-HA reactivity indicates PKA activation. (E) Wdr3 deficiency does not have any effect on Bim induction during ER stress induced by thapsigargin (100 nM) treatment for 24 h. BiP/GRP78 staining was used as a marker for ER stress. Source data are available online for this figure.

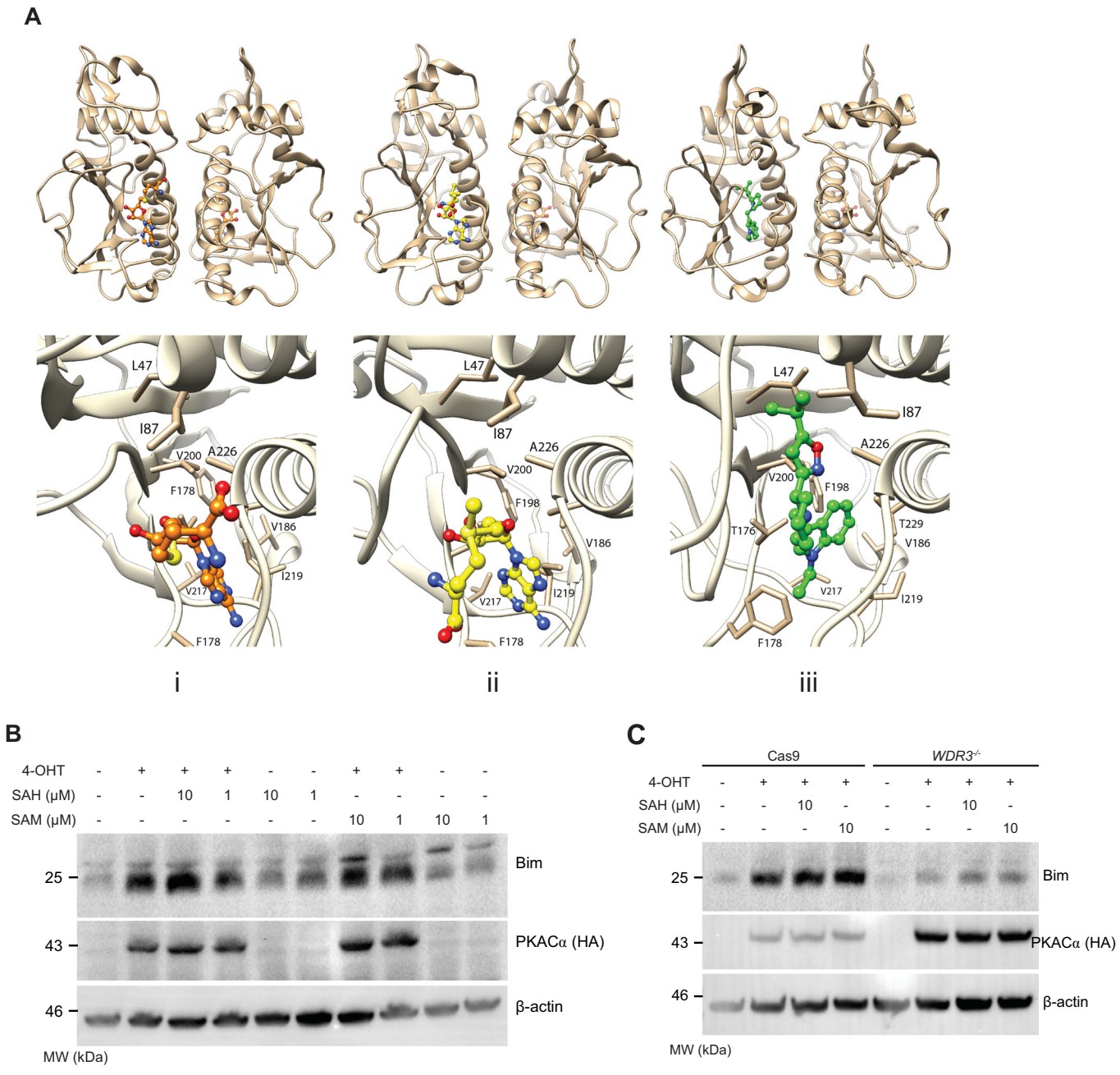

**Figure 5. Molecular modelling of BR43 with Wdr3.**

(A) Upper panel: Ribbon diagram of the dimer of NEP1 domain with (i) SAH, (ii) SAM, and (iii) BR43 bound in one (front/left) site, and with SAH bound in the second (rear) site. Bottom panel: Ligand-binding site of NEP1 domain with (i) SAH, (ii) SAM, and (iii) BR43 bound. Side-chain atoms lining the ligand-binding site are presented in stick representation in tan. Carbon atoms coloured orange (SAH), yellow (SAM) and green (BR43), respectively, with nitrogen, oxygen and sulphur coloured blue, red and yellow, respectively. Figures were produced using the Chimera package (ScienceOpen.com). (B) SAM/SAH could induce Bim under PKA activation. The PKAα MEFs were treated with varying concentrations of SAM or SAH with or without 4-OHT and cell extracts were analysed by Western blot with antibodies as indicated. (C) Bim induction by SAH/SAM under PKA activation is Wdr3-dependent. Parental PKAα MEFs expressing Cas9 or $Wdr3^{-/-}$ MEFs were treated with varying concentrations of SAM/SAH in the presence or in the absence of PKA activation and analysed by Western blots with antibodies as indicated. Source data are available online for this figure.

Finally, our molecular modelling efforts identified that BR43 is capable of binding to the ligand-binding pocket of the NEP1 component of the SSU complex (Thomas et al, 2011), possibly competing with SAM/SAH in inducing Bim. How SAM/SAH binding to this complex induces Bim is yet to be elucidated, however, this study provides a cautionary tale on the widespread use of SAM as a health supplement, i.e. in people with underlying predisposition to systolic heart failure (such as those with hypertension, diabetes, obesity, chronic kidney disease etc.), the use of SAM could seriously complicate the disease outcome.

# Methods

### Reagents and tools table

| Reagent/resource | Reference or source | Identifier or catalogue number |
|---|---|---|
| **Experimental models** | | |
| Adherent HEK293T cells | ATCC | CRL-3216 |
| Mouse embryonic fibroblasts expressing PKACa | Lee et al, 2013 | N/A |
| Rat embryonic heart cells (H9c2) | ATCC | CRL-1446 |
| **Antibodies** | | |
| Anti-Bim 3C5 (1 µg/ml) | Puthalakath et al, 2007 | N/A |
| Anti-phospho-CREB (2 µg/ml) | Cell Signaling | #9198 |
| Anti-acetylated lysine (2µg/ml) | Cell Signaling | #H6908 |
| Anti-HSP70 (0.5 µg/ml) | Sigma-Aldrich | #SAB4200714 |
| Anti-FLAG (1 µg/ml) | Sigma-Aldrich | #F1804 |
| Anti-HA (1 µg/ml) | Sigma-Aldrich | #H6908 |
| Anti-Wdr3 (2 µg/ml) | MyBiosource | #MBS3224353 |
| **Oligonucleotides and other sequence-based reagents** | | |
| qPCR Bim (F) | GGAGATACGGATTGCACAGGAG | N/A |
| qPCR Bim (R) | CTCCATACCAGACGGAAGATAAAG | N/A |
| qPCR GAPDH (F) | CATCACTGCCACCCAGAAGACTG | N/A |
| qPCR GAPDH (R) | ATGCCAGTGAGCTTCCCGTTCAG | N/A |
| Wdr3 CRISPR g1 | TAGCTGATCGTACTTTAAGC | N/A |
| Wdr3 CRISPR g2 | AAGTGGCCTCTACCGTCTAA | N/A |
| Wdr3 CRISPR g3 | AAGTGGCCTCTACCGTCTAA | N/A |
| **Chemicals, enzymes, and other reagents** | | |
| DMEM | Life Technologies | #11885-084 |
| Foetal Calf Serum | Sigma-Aldrich | #12007 C |
| ʟ-glutamine | Sigma-Aldrich | #G7513 |
| Penicillin/Streptomycin | Life Technologies | #15140-148 |
| Trizol | ThermoFisher | #15596026 |
| Affinityscript | Agilent | #600559 |
| ddPCR Multiplex Supermix | Bio-Rad | #12005909 |
| Rolipram | Sigma-Aldrich | #557330-5MG |
| Dobutamine HCl | Sigma-Aldrich | #D0676 |
| Isoproterenol HCl | TOCRIS | #1747 |
| Atenolol | Sigma-Aldrich | #PHR1909 |
| Doxorubicin HCl | Sigma-Aldrich | #D1515-10MG |
| TCEP | Sigma-Aldrich | #C4706-2G |
| Triethyl ammonium bicarbonate | Sigma-Aldrich | #T7408-100ML |

| Reagent/resource | Reference or source | Identifier or catalogue number |
|---|---|---|
| Trypsin | Sigma-Aldrich | #T6567 |
| Acclaim PepMap C18 column | Thermo Fisher | #164946 |
| Nanopore Native Barcoding Kit | Oxford Nanopore Technologies | #SQKNBD114 |
| S-Adenosyl-Methionine | Sigma-Aldrich | #798231 |
| S-Adenosyl-Methionine | Sigma-Aldrich | #1012112-50MG |
| Sodium deoxycholate | Sigma-Aldrich | #6750 |
| Acetonitrile | Sigma-Aldrich | #271004-100 ML |
| **Software** | | |
| VevoLab v3.1.0 | Visualsonics, CA | N/A |
| Proteome Discoverer v2.4 | Thermo Scientific, Bremen, Germany | N/A |
| NanoPlot v1.42.0 | De Coster and Rademakers, 2023 | N/A |
| Filtlong v0.2.1 | https://github.com/rrwick/Filtlong | N/A |
| YASARA | Land and Humble, 2018 | N/A |
| AMBER force field | Maier et al, 2015 | N/A |
| AutoDock VINA | https://vina.scripps.edu/ | N/A |
| MolAlign | Brown et al, 2019 | N/A |
| **Other** | | |
| BCA protein assay kit | Pierce | #23228 |
| NuPAGE™ 4–20%, bis-tris 1.0 mm, mini protein gels, 10 wells | Invitrogen | #NP0321BOX |
| Orbitrap Eclipse™ Tribrid™ Mass Spectrometer | Thermo Scientific | FSN04-10000 |
| Mini osmotic pumps | Alzet osmotic pumps | #2001D |

## Preparation of experimental models and subject details

### *Cell lines, culture conditions*

Mouse embryonic fibroblasts (MEFs) and the rat embryonic cardiomyocyte cell line (H9C2) were cultured in Dulbecco's Modified Eagle Medium (DMEM, Life Technologies Cat #11885-084)) supplemented with 10% foetal calf serum (FCS; Sigma Cat #12007 C), 2 mM ʟ-glutamine (Sigma Cat #G7513 and 1% Penicillin/Streptomycin (Life Technologies Cat #15140-148) and incubated at 37 °C under 10% $CO_2$. 4-OHT-inducible MEFs expressing the catalytic subunit of PKA (PKACα) has been described before (Lee et al, 2013) and were induced with 5 nM 4-Hydroxytamoxifen.

## Antibodies

Anti-Bim antibody (3C5; 1 µg/ml) was a kind gift from Andreas Strasser (WEHI). Anti-phospho-CREB antibody (Cat # 9198; 1:1000) and anti-acetylated lysine antibody (Cat #9441; 1:1000) were

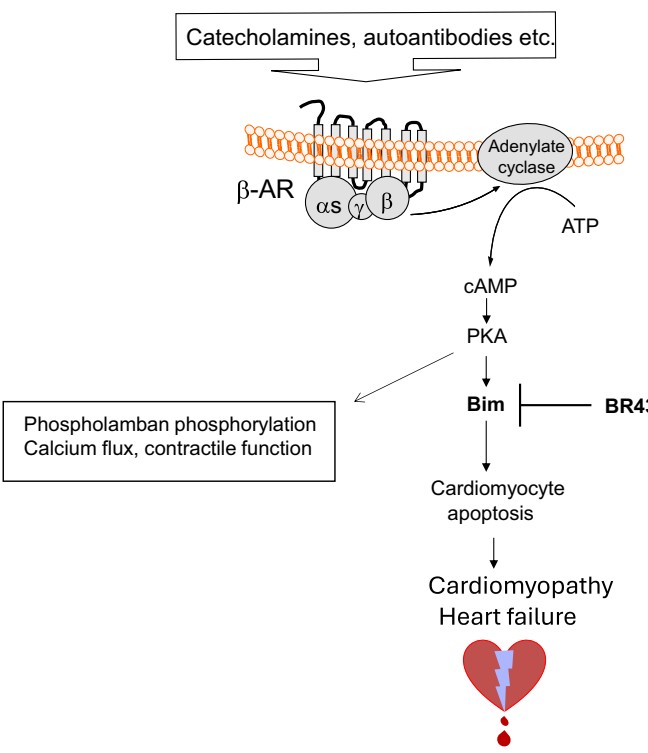

**Figure 6. Proposed model of BR43 action on heart muscle cells.**

β-adrenergic receptor activation at the cell surface leads to the activation of adenylate cyclase, which produces cAMP. cAMP activates PKA, which is necessary for the contractile function. Dysregulated β-AR activation induces Bim expression and apoptosis. Selective inhibition of Bim expression by BR43 maintains PKA activation and the contractile function and therefore is hemodynamically neutral.

purchased from Cell Signaling Technology (Danvers, MA, USA). Anit-HA (Cat #H6908; 0.8 µg/ml), anti-HSP70 (Cat #SAB4200714; 1:2000) and anti-FLAG (Cat #F1804; 1:5000) were purchased from Sigma-Aldrich (Merck, Macquarie Park, NSW 2113, Australia). Anti-Wdr3 middle region antibody (Cat #MBS3224353; 1:1000) was purchased from MyBiosource (San Diego, CA, USA).

## RNA analysis

RNA quantitation was carried out by droplet digital PCR (ddPCR) using QX200™ Droplet Digital PCR system (Bio-Rad) or by real-time quantitative PCR using magnetic induction cycler (MIC) system (Bio Molecular Systems). RNA was isolated using Trizol™ (Thermo Fisher Cat #15596026) and converted to cDNA using Affinityscript™ cDNA synthesis kit (Agilent Cat #600559) following the manufacturer's instructions. Equal quantities of RNA were taken for cDNA synthesis, and total transcripts per µg of total RNA was calculated in the droplet digital system, and relative quantitation using cycle threshold ($C_t$) was used in real-time qRT-PCR analysis. GAPDH was used as the housekeeping gene in qRT-PCR analysis.

## Drug screening

A high-throughput amenable assay was developed and utilised to run a screen of 14 × 384-well microtitre plates totalling ~5000

WECC (wide range external control compound) library compounds. In brief, rat embryonic cardiomyocyte cells (H9c2) were seeded into 384-well tissue culture plates (1000 cells/well) in DMEM + 10% FCS. Twenty-four hours post-seeding, media was replaced with serum-free media and the cells were treated with 1 µM of the phosphodiesterase inhibitor Rolipram as well as an EC70 dose of the beta-adrenergic receptor agonist, Dobutamine and 10 µM of ~5000 WECC library compounds. Twenty-four hours post-treatment, cell viability was assessed via an Alamar blue assay, and fluorescence was quantified on an EnVision plate reader.

## Medicinal chemistry and drug synthesis

See the appendix for the details of chemical synthesis.

## Animal experiments

Less than 12 weeks old C57B/6 mice (equal number of males and females) were used for all the animal experiments. For the safety study, mice were injected intraperitoneally with 0.79 mg/kg body weight of BR43 or the equivalent volume of DMSO carrier daily for 7 days. Mice were monitored for weight, and after 7 days, mice were culled, blood and organs were collected for analyses. The work was carried out under La Trobe University's animal ethics approval no. AEC 19013.

For the efficacy study, baseline heart parameters were measured one week before the start of drug treatment. Isoproterenol (1.25 mg/kg/h) or vehicle was administered via implanted osmotic minipump (Model 2004, Alzet Osmotic Pumps, Cupertino, CA 95014-4166, USA) for 24 h until removal. BR43 (0.3 mg/kg) or vehicle control were injected (IP) daily for 7 days. On day 7, echocardiography studies were performed using an ultra-high-frequency preclinical ultrasound system (Vevo 2100, Visualsonics, CA) equipped with a 40 MHz linear array transducer to acquire all imaging while under isoflurane (1.7%) anaesthesia. Images were analysed offline using the manufacturer's official software (Vevo-Lab v3.1.0, Visualsonics, CA). The animals were subsequently euthanised for sample collection. This work was carried out under the Baker Institute animal ethics approval no. E/1962/2019/B. For the toxicity study, the mice were injected with doxorubicin 2.5 mg/kg (i.p), every other day for 12 days for a total cumulative dose of 15 mg/kg. BR43 (0.3 mg/kg) or atenolol (0.5 mg/kg) were also co-administered for the duration, and serum samples were collected for troponin analysis by ELISA on day 13. This work was carried out under the LTU ethics approval AEC25002. All animal experiments were conducted in a double-blinded format.

## Thermal proteome profiling

### Sample preparation for proteomic analysis

Sample preparation for thermal proteome profiling was carried out as previously described (Savitski et al, 2014). Equal volumes from the ten aliquots heated to different temperatures in each experiment was reduced using TCEP (tris(2-carboxyethyl) phosphine) at a final concentration of 5 mM and alkylated using iodoacetamide at a final concentration of 20 mM. The reduced and alkylated proteins were precipitated by adding four volumes of ice-cold acetone and by incubating at −80 °C for 4 h. The tubes were then centrifuged at 21,000 × g for 15 min, and the acetone was discarded. The protein pellets were

air-dried until the acetone odour was removed. About 100 µl of digestion buffer consisting of trypsin and 20 mM triethyl ammonium bicarbonate was added, and digestion was carried out at overnight on a shaker set to 37 °C. The ratio of trypsin:protein was maintained at 1:50. The digested peptides were labelled with 10-plex TMT (tandem mass tags) as per the manufacturer's protocol. Briefly, TMT vials were reconstituted with anhydrous acetonitrile, and the reagent was mixed with peptides. The labelling reaction was carried out for 1 h at room temperature and was quenched by adding excess hydroxylamine. In each experiment, increasing temperatures were labelled with increasing mass TMT tags, i.e. 37, 41, 44 °C… were labelled with 126, 127N, 127C, etc. Peptides from each experiment were pooled and fractionated by high pH reverse phase liquid chromatography using spin columns into 12 fractions as described previously (Liu et al, 2023). The eluted fractions were evaporated in a speed vac and reconstituted in mass spec buffer (2% acetonitrile, 0.1% trifluoroacetic acid in water).

### Mass spectrometry data acquisition

The fractionated TMT-labelled peptides were analysed on an Orbitrap Eclipse mass spectrometer interfaced with an Ultimate 3000 nano UHPLC system. Each fraction was first loaded onto a trap column (PepMap $C_{18}$ 100 µm ID × 2 cm length trap column, Thermo Fisher Scientific) and washed for 6 min at 12 µl/min. The trap column was then brought in line with the analytical column (BEH $C_{18}$, 1.7-µm particle size, 130 Å pore size and 75 µm ID × 25 cm length, Waters). Separation of peptides was performed at 55 °C, 250 nL/minute, using a linear gradient of buffer A (0.1% formic acid, 2% ACN, in water) and buffer B (0.1% formic acid, 80% ACN, in water), starting from 2% B to 35% B over 90 min. Highly hydrophobic peptides were eluted by increasing buffer B to 50% over the next 15 min. The column was washed by increasing buffer B to 95% over the next 5 min and holding at 95% for 5 min. The percentage of buffer B was dropped to 1% in 2 min, and the column was equilibrated for 3 min. The total runtime was 120 min.

The Orbitrap Eclipse was operated in the data-dependent acquisition (DDA) mode in positive polarity. The Synchronous Precursor Selection—Real Time Search—MS3 (SPS—RTS— MS3) method was employed for data acquisition to enable fast and highly accurate TMT-based quantitation (Erickson et al, 2019; Schweppe et al, 2020). The UniProt *Mus musculus* canonical protein database (downloaded March 2023) was utilised for RTS. Other mass spec settings are given below:

MS1 –> Orbitrap detector, Resolution = 120 K, Scan range = 400–1600, max injection time = 50 ms, AGC target = 400,000.

MS2 –> Ion Trap detector, Scan rate = Turbo, max injection time = 35 ms, AGC target = 10,000.

MS3 –> Orbitrap detector, Resolution = 50 K, Scan range = 100-500, max injection time = 200 ms, AGC target = 100,000.

### Database searches and bioinformatics analysis

Mass spectrometry data processing was carried out as described previously (Datta et al, 2021). Raw files were searched against the *Mus musculus* UniProt canonical protein database using Sequest HT through Proteome Discoverer (Version 2.4) (Thermo Scientific, Bremen, Germany). Precursor and fragment mass tolerance were set to 10 ppm and 0.05 Da, respectively. TMT at the N-terminus and lysine and carbamidomethylation of cysteine were set as fixed modifications, while oxidation of methionine was set as a dynamic modification. A false discovery rate (FDR) threshold of 1% was used to filter peptide spectrum matches (PSMs). FDR was calculated using a decoy search. PSMs were quantified based on TMT-reporter ion intensities and were rolled up into proteins using the "Reporter Ions Quantifier" node of Proteome Discoverer. The protein level quant data was exported as spreadsheets and were used for downstream analysis.

Data analysis was carried out as detailed previously (Franken et al, 2015). Briefly, the "TPP-TR" (temperature range) R-package was executed in RStudio to normalise, filter, detect intersections between datasets, compute fold changes, and fit melt curves. Only those proteins that were identified as putative drug targets by the R-package were considered for further experimental validation.

### CRISPR knockout cell lines

CRISPR knockout cell lines were generated using the lentiviral system that was described by Aubrey et al (Aubrey et al, 2015). The following guides spanning three different regions were used for generating the knockouts g1: TAGCTGATCGTACTTTAAGC; g2: AAGTGGCCTCTACCGTCTAA and g3: AAGTGGCCTCTACCG TCTAA. Lentiviral particles with all three guides were mixed in equal proportion before infecting the target cells. Knockout clones were identified by nanopore sequencing. The CRISPR target regions were amplified from individual clones by PCR, and libraries were prepared for Oxford Nanopore sequencing using the Native Barcoding Kit (SQK-NBD114) and sequenced on a PromethION 2 Solo using a PromethION flow cell (R10). Basecalling was performed on-device in super-accurate mode. Reads were inspected with NanoPlot v1.42.0 (De Coster and Rademakers, 2023) and filtered using Filtlong v0.2.1 (https://github.com/rrwick/Filtlong).

### Molecular modelling

Small molecule ligands were docked to proteins using the VINA software (Trott and Olson, 2010); protein structures were obtained from the PDB. Docking of BR43 to the WD repeat-containing protein 3 (Wdr3) component of the human small-subunit processome (PDBid 7MQA, Y chain (Singh et al, 2021)) identified high-scoring docking poses with the ligand bound to the first beta-propeller domain; subsequent MD simulations with ligand bound included only this domain, residues K5-G17, S416-E526 and R537-T705. Docking and MD simulations to the dimeric ribosomal RNA small-subunit methyltransferase NEP1 component (chains GB and HB) included residues R41–I244 in both monomers.

Subsequent to small-molecule ligand docking, hydrogen atoms were added to fill valencies (with ionisable groups in their standard state at a pH of 7.4), the system solvated in a cubic water box extending at least 5 Å beyond all atoms, and $K^+$ and $Cl^-$ ions added to neutralise the cell ionic and deliver an ionic mass density of 0.9%, using the YASARA software (Land and Humble, 2018). Structures were then minimised utilising the AMBER force field (Maier et al, 2015), fixing all bonds and angles to hydrogen atoms. Molecular dynamics (MD) simulations were then run for 250 ns in the NPT ensemble using periodic cell boundaries, a time step of 1.0 fs, at a temperature of 298 K maintained with velocity rescaling (rescaled every ten simulation steps) and pressure controlled by keeping the water density at 0.997 g/ml. Long-range electrostatic

**Paper explained**

**Problem**

Current treatments for systolic heart failure, which aim to improve heart contractility and cardiac output, face several challenges. Many therapies, such as inotropes, temporarily boost contractility but often increase the risk of arrhythmias and mortality with long-term use. Beta-blockers and ACE inhibitors improve survival but may not sufficiently restore contractile function in advanced cases. Additionally, some patients experience adverse side effects or have contraindications limiting treatment options. Device therapies like implantable defi-brillators address arrhythmias but do not directly enhance contractility. Overall, existing treatments often provide symptomatic relief without fully reversing impaired myocardial function, highlighting a critical need for safer, more effective therapies.

**Results**

The paper explains a novel strategy to develop a hemodynamically neutral drug-like compound. A high-throughput screening identified several drug-like compounds that reversed b-AR-mediated cell death in the rat cardiomyocyte cell line H9c2. These compounds were validated in a system to identify compounds that will selectively inhibit b-AR-mediated apoptosis (those down regulate Bim) but not affect PKA activity, which is needed for heart contractility. Structure-activity relationship analysis (SAR) led to a compound with minimal cytostatic effect and found to be safe in mice in dose escalation studies. The compound was also found to be effective in reversing cardiomyopathy in the takotsubo cardiomyopathy model without compromising heart contractility as well as in reversing drug-induced cardiotoxicity. Thermal proteome profiling (TPP) mass spectrometry identified protein Wdr3 as a possible drug target, which was confirmed by CRISPR knockout studies.

**Impact**

The discovery of a drug-like compound that reverses cardiomyopathy without altering cardiac contractility or output represents a major therapeutic breakthrough, offering disease modification without hemodynamic compromise. This approach may significantly improve patient outcomes, reduce reliance on current high-risk treatments and redefine the management of heart failure at its molecular root.

forces were simulated using the particle-mesh Ewald method. Following MD, the structures were minimised for further analysis. Small-molecule binding affinities between protein and ligand (i.e. with all water and ions removed) were estimated using the VINA software with flexible receptor and ligand.

MD simulations of the dimeric NEP1 component included in one monomer a single molecule of SAH in the ligand-binding site in addition to the ligand of interest (i.e. either BR43, SAH or SAM).

Ligand replacement was performed using the MolAlign web server (Brown et al, 2019). Following molecular superposition, protein-ligand complexes were subject to MD simulation and binding affinity estimation as described above.

# Data availability

The proteomic data (thermal proteome profiling mass spectrometry) is available in The PRoteomics IDEntification (PRIDE) database. Project accession: PXD065822, Token: zdfxITC9totq.

The source data of this paper are collected in the following database record: biostudies:S-SCDT-10_1038-S44321-025-00284-6.

# Peer review information

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

## Acknowledgements

We would like to thank Helen Kiriazis for the help with Echocardiograms; L Moorfield and the Office of Alumni and Advancement for facilitating the financial contribution from HP for this project; Rohan Lowe and the Proteomics Research Platform, La Trobe University, for the mass spectrometry work; R Anders and J Goding for critically reviewing the manuscript.

## Author contributions

**Corey Pollock**: Investigation. **Xilun Wang**: Investigation. **Hussam Alsaraji**: Investigation. **Joseph Menassa**: Investigation; Conducting experiments. **George Mbogo**: Investigation. **Dimuthu Angage**: Investigation. **Benjamin Richards**: Investigation. **Jason Glab**: Investigation. **Keshava K Datta**: Investigation. **Liana Theodoridis**: Investigation; Methodology. **Steve Petrovski**: Investigation; Methodology. **Daniel Donner**: Investigation. **Yuvixza Lizarme-Salas**: Investigation; Methodology. **Xiao-Jun Du**: Data curation; Supervision; Investigation. **Michael Foley**: Formal analysis; Writing—review and editing. **Brian J Smith**: Software; Validation; Investigation; Methodology. **Belinda**

**Abbott**: Conceptualisation; Data curation; Supervision; Investigation; Methodology; Writing—review and editing. **Hamsa Puthalakath**: Conceptualisation; Data curation; Supervision; Investigation; Writing—original draft; Project administration; Writing—review and editing.

Source data underlying figure panels in this paper may have individual authorship assigned. Where available, figure panel/source data authorship is listed in the following database record: biostudies:S-SCDT-10_1038-S44321-025-00284-6.

## Disclosure and competing interests statement

The authors declare no competing interests.

# Expanded View Figures

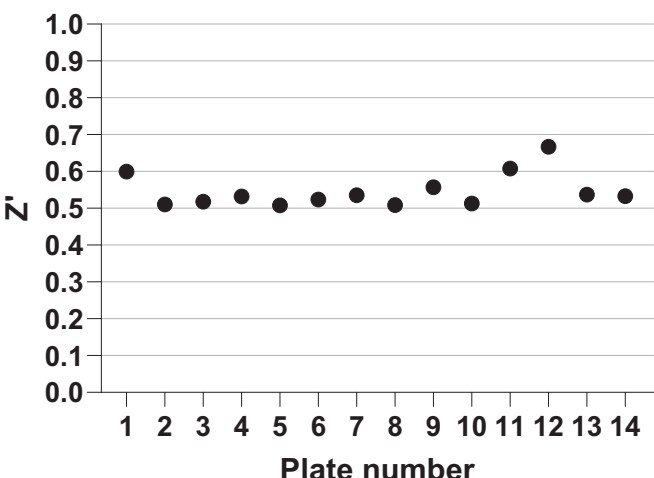

**Figure EV1. The high-throughput screen.**

The HTS shows that Z' value exceeded 0.4 for all 14 assay plates. The Z' value provides an indication of how robust a HTS assay is, whereby Z' > 0.4 (dashed line) indicates cell-based assay robustness and suitability for HTS.

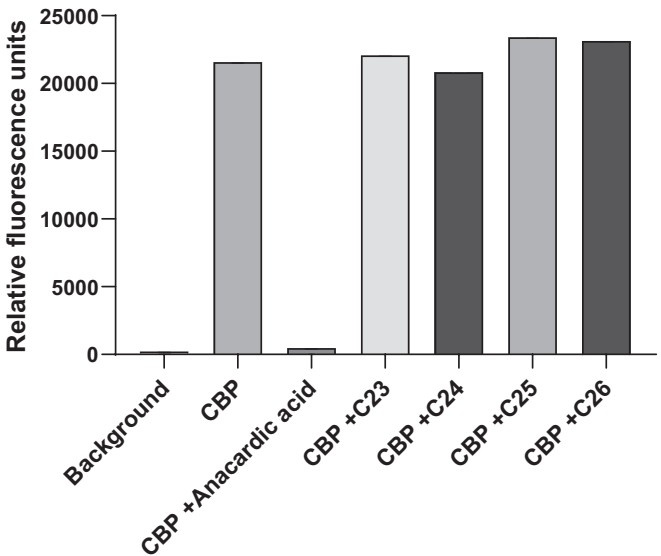

**Figure EV2. Histone acetyltransferase (HAT) assay showing the drug-like compounds did not have HAT-inhibitor activity.**

The HAT assay was performed using a HAT assay kit (Cat #56100, Active Motif, Carlsbad, CA 92008, USA), using HAT-inhibitor anacardic acid as a control. Each compound was tested at 10 µM concentration as per manufacturer's instructions. Source data are available online for this figure.

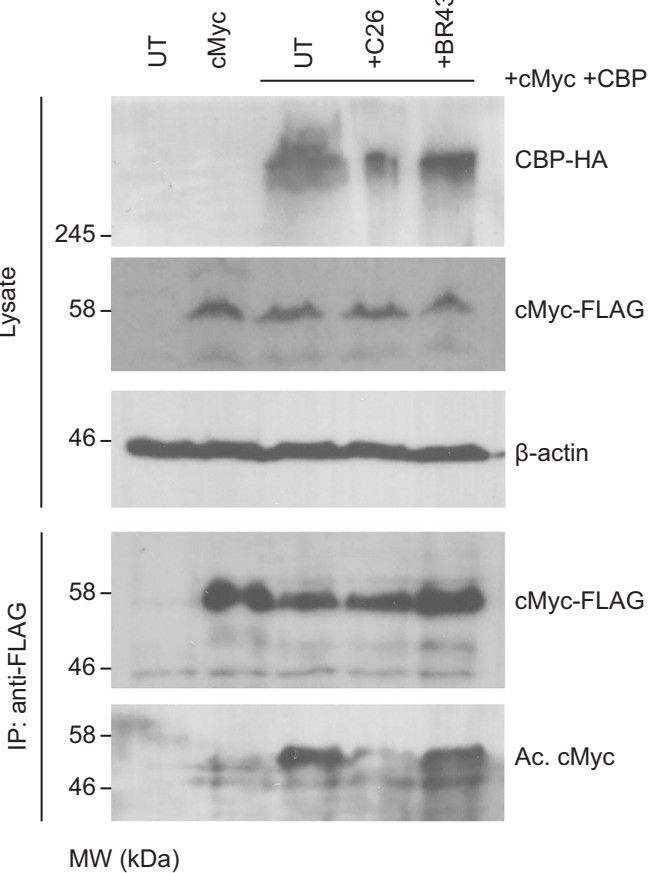

**Figure EV3. The drug-like compounds do not inhibit c-Myc acetylation by CBP.**

HEK293T cells were transfected with FLAG-c-Myc with or without HA-tagged CBP. Cells co-expressing CBP and c-Myc were left either untreated or treated with 10 mM of C26 or BR43 and analysed c-Myc was pulled down with anti-FLAG beads and analysed by Western blots with anti-FLAG and with anti-acetyl lysine antibodies. Source data are available online for this figure.

**Figure EV4.  Synthesis of C26.**

The convergent strategy involved N-alkylation of 3-(pyrrolidine-2-yl) isoxazole (2) using bromomethylindole (1) to yield compound 3. This was subsequently subjected to Boc-deprotection to yield compound 4, which was N-alkylated further using ethyl bromide to get C26.

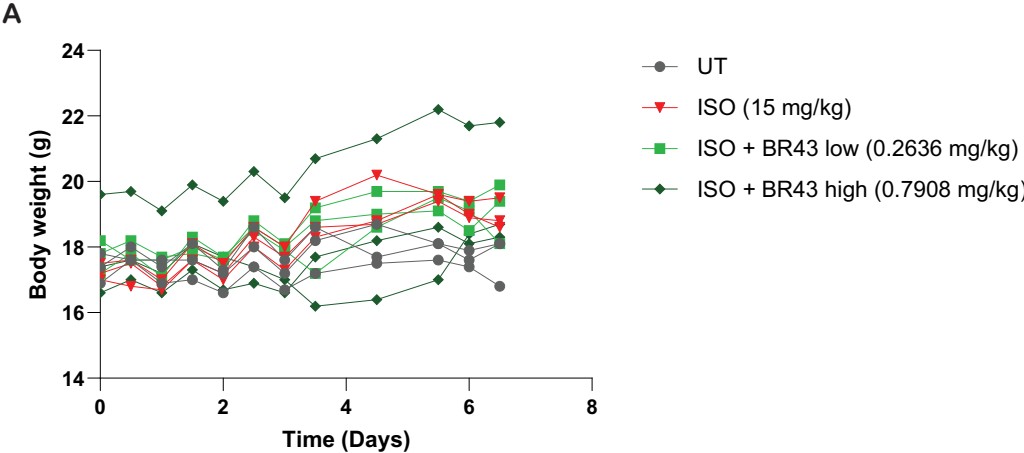

B

| | Heart | Kidney | Liver | Spleen |
|---|---|---|---|---|

Control

BR-43

C

RBC

HB

Ht

RDW

MCH

MCV

Plt

◀ **Figure EV5. In vivo safety studies of BR43.**

(A) C57B/6 were given PBS (UT control) or isoproterenol (ISO) using Alzet minipumps for a period of 24 h and subsequently given BR43 (I.P) for seven consecutive days and body weight was monitored. (B) H&E staining of various tissues at the end of 7 days and comparisons were made between control (UT) and Iso+BR43 high (0.79 mg/kg). (C) Various dozes of BR43 were injected (I.P) for 7 consecutive days and blood samples were analysed for various blood parameters as shown. Error bars ± SD, $n = 4$-8 animals in each group, one-way ANOVA with Tukey's multiple comparison test. Source data are available online for this figure.

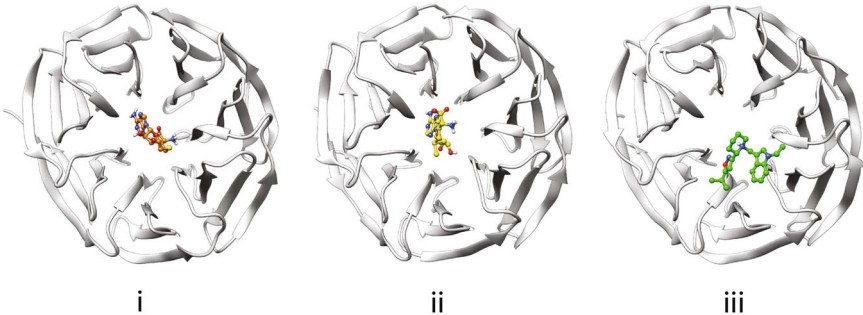

i    ii    iii

**Figure EV6.  SAH and SAM could induce Bim expression through Wdr3.**

Ribbon diagram of the first beta barrel domain of Wdr3 complexed with (i) SAH, (ii) SAM and (ii) BR43. Ligands are presented in ball-and-stick representation, with carbon atoms coloured orange (SAH), yellow (SAM) and green (BR43), respectively, with nitrogen, oxygen and sulphur coloured blue, red and yellow, respectively.

