## [Peer Review File · EMBO Molecular Medicine]

Development of a novel therapy for systolic heart failure

Hamsa Puthalakath, Corey Pollock, Xilun Wang, Hussam Alsaraji, Joseph Menassa, George Mbogo, Dimuthu Angage, Benjamin Richards, Jason Glab, Keshava Datta, Liana Theodoridis, Steve Petrovski, Daniel Donner, Yuvixza Lizarme-Salas, Xiao-Jun Du, Michael Foley, Brian Smith, and Belinda Abbott

Corresponding authors: Hamsa Puthalakath (h.puthalakath@latrobe.edu.au) , Belinda Abbott (b.abbott@latrobe.edu.au)

Review Timeline:

Submission Date:	10th Dec 24
Editorial Decision:	22nd Jan 25
Revision Received:	2nd Jun 25
Editorial Decision:	20th Jun 25
Revision Received:	15th Jul 25
Accepted:	17th Jul 25

Editor: Zeljko Durdevic

Transaction Report:

22nd Jan 2025

Dear Dr. Puthalakath,

Thank you for the submission of your manuscript to EMBO Molecular Medicine, and please accept my apologies for the unusual delay in getting back to you. We have now received feedback from two of the three reviewers who agreed to evaluate your manuscript. As the referee #3 will unfortunately not be able to return his/her report in a timely manner, and given that both reviewers provide similar recommendations, we prefer to make a decision now in order to avoid further delay in the process. Should referee #3 provide a report, we will send it to you, with the understanding that we will not ask for an additional revision.

As you will see from their reports pasted below, both referees recognize potential interest of the manuscript but also raise important concerns that should be addressed in a major revision. Confirmation of the main findings in an additional animal model as suggested by the referee #2 should be the focus of the revision. Given that the revision will require extensive experimentation we think six months rather than three months would be more appropriate to provide the complete revision. If you would like to discuss further the points raised by the referees, I am available to do so via email or video. Let me know if you are interested in this option.

We would welcome the submission of a revised version within six months for further consideration. Please let us know if you require longer to complete the revision.

I look forward to receiving your revised manuscript.

Yours sincerely,

Zeljko Durdevic

We require:

- 1) A .docx formatted version of the manuscript text (including legends for main figures, EV figures and tables). Please make sure that the changes are highlighted to be clearly visible.
- 2) Individual production quality figure files as .eps, .tif, .jpg (one file per figure). For guidance, download the 'Figure Guide PDF': (<https://www.embopress.org/page/journal/17574684/authorguide#figureformat>).
- 3) A .docx formatted letter INCLUDING the reviewers' reports and your detailed point-by-point responses to their comments. As part of the EMBO Press transparent editorial process, the point-by-point response is part of the Review Process File (RPF), which will be published alongside your paper.
- 4) A complete author checklist, which you can download from our author guidelines

(<https://www.embopress.org/page/journal/17574684/authorguide#submissionofrevisions>). Please insert information in the checklist that is also reflected in the manuscript. The completed author checklist will also be part of the RPF.

6) It is mandatory to include a 'Data Availability' section after the Materials and Methods. Before submitting your revision, primary datasets produced in this study need to be deposited in an appropriate public database, and the accession numbers and database listed under 'Data Availability'. Please remember to provide a reviewer password if the datasets are not yet public (see <https://www.embopress.org/page/journal/17574684/authorguide#dataavailability>).

.

- the medical issue you are addressing,

- the results obtained and

- their clinical impact.

12) Author contributions: You will be asked to provide CRediT (Contributor Role Taxonomy) terms in the submission system. These replace a narrative author contribution section in the manuscript.

13) A Conflict of Interest statement should be provided in the main text.

14) Every published paper now includes a 'Synopsis' to further enhance discoverability. Synopses are displayed on the journal webpage and are freely accessible to all readers. They include a short stand first (maximum of 300 characters, including space) as well as 2-5 one-sentences bullet points that summarizes the paper. Please write the bullet points to summarize the key NEW findings. They should be designed to be complementary to the abstract - i.e. not repeat the same text. We encourage inclusion of key acronyms and quantitative information (maximum of 30 words / bullet point). Please use the passive voice. Please attach these in a separate file or send them by email, we will incorporate them accordingly.

15) Include a Reagents and Tools Table as part of the Methods section, which can be downloaded from our author guidelines (<https://www.embopress.org/page/journal/17574684/authorguide#structuredmethods>)

***** Reviewer's comments *****

Referee #1 (Comments on Novelty/Model System for Author):

This paper presents a novel hemodynamically neutral therapy for heart failure. By establishing cell models, conducting high-throughput screening, and refining the structure-activity relationship, the lead compound BR43 was identified. Through a series of validations, including in vitro experiments, animal models, and target identification, BR43 was proposed as a promising candidate for treating heart failure, offering potential improvements over current clinical therapies. This work demonstrates strong innovation and potential, with significant practical value for heart failure treatment.

However, the paper still has some shortcomings: 1. Although the authors suggest that the entire signaling pathway is related to apoptosis, specific cellular and biochemical phenotypes should be provided to clearly demonstrate how the drug induces apoptosis; 2. In the cell proliferation experiments, it would be helpful to include more concentration points, plot dose-response curves, and calculate the IC50 values. This would allow for a more direct comparison of the drug's safety profile. Additionally, it should be clarified whether the observed phenotype is indeed due to apoptosis. 3. In the animal models, there is a lack of evaluation using relevant biomarkers of heart failure, such as BNP, which would help to clearly demonstrate the effect of BR43 on the disease improvement. 4. It would be beneficial if the authors could present more detailed results from mass spectrometry and provide a more thorough description of the process used to screen candidate proteins. 5. Given the challenges in performing affinity experiments for Wdr3, it would be helpful for the authors to use additional computational methods to predict the binding affinity between BR43 and Wdr3. 6. It is noted that there are few references from the past five years in the manuscript. The authors are encouraged to add more recent references to ensure the paper is up-to-date.

This paper provides a comprehensive account of the discovery and modification of a compound, along with in vitro and in vivo experiments, proposing BR43 as a candidate for treating heart failure. The experimental results suggest that BR43 has promising clinical application potential and offers valuable theoretical and practical guidance for drug development in heart failure.

Referee #2 (Comments on Novelty/Model System for Author):

This is a novel study, aiming at identifying new compounds to treat heart failure. The authors have employed a number of techniques and convincingly demonstrated BR43 as a candidate drug to treat heart failure.

The reasons of my less enthusiastic recommendation are the follows.

1-The in vivo model is isoproterenol injection. Whereas it is appropriate, it should not be the only one. I would recommend additional mouse models, including genetic models to be tested.

2-Data quality. Some immunoblots should be improved and quantified.

Referee #2 (Remarks for Author):

In this study, Pollock et al. focused on the identification of novel compounds to treat heart failure. The authors started with screening for similar or better inhibitors of beta-AR signaling and cell death in H9c2 cells. They also investigated the effects on

PKA signaling and Bim expression. They next spent many efforts to narrow down and refine the candidate compounds. In the end, they generated a modified version of their leading compounds. In vivo tests showed the this BR43 compound significantly improved isoproterenol-induced heart failure and also inhibited the expression of Bim.

1. The authors started with beta-AR signaling as the foundation for screening. However, later, they more focused on beta-AR-independent signaling, which is likely better considering the plethoric effect of inhibiting beta-AR in vivo. This is interesting. However, the in vivo test still used isoproterenol in mice. What about the therapeutic effects of BR43 in conventional heart failure model, such as transverse aortic constriction?

2. Along the similar line as in (1), it would be better if the authors could conduct a side-by-side comparison to delineate the effects of BR43 as opposed to other mainstream drugs in vivo.

3. Some immunoblots are suboptimal. In addition, quantification should be conducted for all immunoblots.

4. In addition to H9c2 cells, primary neonatal or adult myocytes should be employed. If possible, human iPSC-Cardiomyocytes may be used.

Dear Editors,

Thank you for reviewing our manuscript and thank you for the valuable and constructive feedback from the reviewers. We would also like to thank the editors for giving us enough time to conduct the additional animal experiments (obtaining ethics approval from the relevant bodies has its own challenges and we appreciate the editors taking this into consideration). We have addressed all the reviewers' comments as given below:

Reviewer 1:

1. Although the authors suggest that the entire signaling pathway is related to apoptosis, specific cellular and biochemical phenotypes should be provided to clearly demonstrate how the drug induces apoptosis.

Answer: There must be some misunderstanding. The drug does not induce apoptosis. It blocks apoptosis by reducing b-AR-mediated Bim induction. How Bim induces apoptosis is well documented by us and various others (Puthalakath et al, Cell 129 (7), 1337-1349; Puthalakath et al, Molecular cell 3 (3), 287-296).

2. In the cell proliferation experiments, it would be helpful to include more concentration points, plot dose-response curves, and calculate the IC50 values. This would allow for a more direct comparison of the drug's safety profile. Additionally, it should be clarified whether the observed phenotype is indeed due to apoptosis.

Answer: We have done IC50 measurements, and it is part of Figure 2C in the revised manuscript. Again, the observed phenotype is due to apoptosis inhibition rather than due to apoptosis.

3. 3. In the animal models, there is a lack of evaluation using relevant biomarkers of heart failure, such as BNP, which would help to clearly demonstrate the effect of BR43 on the disease improvement.

Answer: In the revised manuscript, we have used an additional animal model, and we measured serum troponin levels (a marker for heart failure). It is part of Figure 3 i.e., 3D.

4. 4. It would be beneficial if the authors could present more detailed results from mass spectrometry and provide a more thorough description of the process used to screen candidate proteins.

Answer: Very detailed description of the mass spectrometry analyses is given in the methods section under three subheadings: 1. *Sample preparation for proteomic analysis*, 2. *Mass spectrometry data acquisition* and 3. *Database searches and bioinformatics analysis*.

5. Given the challenges in performing affinity experiments for Wdr3, it would be helpful for the authors to use additional computational methods to predict the binding affinity between BR43 and Wdr3.

Answer: The "Molecular modelling" section of the Results provides the results of our investigations into the binding affinity between BR43 and Wdr3; in short, the ligands were first docked with the receptor, then molecular dynamics simulations conducted prior to estimating the binding affinity. These are all described in the "Molecular modelling" section of the Methods section.

6. It is noted that there are few references from the past five years in the manuscript. The authors are encouraged to add more recent references to ensure the paper is up-to-date.

Answer: Thanks for pointing this out. There is lack of research into developing hemodynamically neutral drugs for treating HFrEF. We had mentioned a few recent attempts such as Sacubitril/Valsartan and SGLT-2. However, we inadvertently omitted Omecamtiv Mecarbil (OM). Though it was touted to be a hemodynamically neutral therapy by activating myosin, FDA approval was denied due to lack of clinical efficacy in 2024. This has been added in the discussion in the revised manuscript.

Reviewer 2:

1. The authors started with beta-AR signaling as the foundation for screening. However, later, they more focused on beta-AR-independent signaling, which is likely better considering the plethoric effect of inhibiting beta-AR in vivo. This is interesting. However, the in vivo test still used isoproterenol in mice. What about the therapeutic effects of BR43 in conventional heart failure model, such as transverse aortic constriction?

Answer: We have conducted additional animal experiments i.e., doxorubicin-induced cardiotoxicity model and this data is included in the revised manuscript (Figure 3D).

2. Along the similar line as in (1), it would be better if the authors could conduct a side-by-side comparison to delineate the effects of BR43 as opposed to other mainstream drugs in vivo.

Answer: Indeed, we have compared BR43 side by side with a clinically used beta blocker atenolol and this data is included in the revised manuscript (Figure 3D). We also had compared BR43 with pindolol in Fig. 1C and Fig. 2D.

3. 3. Some immunoblots are suboptimal. In addition, quantification should be conducted for all immunoblots.

Answer: Figure 3C was pixelated on magnification. This error has been fixed with a new panel with higher resolution. We have also quantified all the immunoblots using ImageJ but due to the crowded nature of the figure panels, we have submitted a separate spreadsheet with the values and figures.

4. 4. In addition to H9c2 cells, primary neonatal or adult myocytes should be employed. If possible, human iPSC-Cardiomyocytes may be used.

Answer: Indeed, we did not rely only on H9C2 cells. Figure 3C is using mouse cardiac tissues (primary adult myocytes).

20th Jun 2025

Dear Dr. Puthalakath,

Thank you for the submission of your revised manuscript to EMBO Molecular Medicine. I am pleased to inform you that we will be able to accept your manuscript pending the following final amendments:

1) Authors:

- E-mail correspondence to Keshavakrishna Datta could not be delivered. Please update the e-mail address and make sure to enter correct e-mail addresses for all authors in our submission system.

- We also note name discrepancy in our submission system and in the manuscript. Keshava Datta in the manuscript and Keshavakrishna Dattain our system. Please correct.

2) Author checklist: Please submit a complete checklist. <https://www.embopress.org/pb-assets/embo-site/EMBO%20Press%20Author%20Checklist-1642513524327.xlsx>

3) In the main manuscript file, please do the following:

- Please address all comments suggested by our data editors listed below:

o Data availability statement:

1. Please note that the data availability statement is not provided in the manuscript.

o Figure legends:

1. Please note that the exact p values are not provided in the legends of figures 1H, 2C, 3A, D.

2. Please indicate the statistical test used for data analysis in the legends of figures 1H, 2C, EV5 C.

3. Please note that information related to n is missing in the legend of figure EV5 C.

4. Although 'n' is provided, please describe the nature of entity for 'n' in the legends of figures 1C, H; 2C, 3A, D.

5. Please note that the error bars are not defined in the legend of figure EV5 C.

- Rename "Competing interest" to "Disclosure and competing interests statement". We updated our journal's competing interests policy in January 2022 and request authors to consider both actual and perceived competing interests. Please review the policy <https://www.embopress.org/competing-interests> and update your competing interests if necessary.

- Author contributions: Please remove it from the manuscript and specify author contributions in our submission system. CRediT has replaced the traditional author contributions section because it offers a systematic machine-readable author contributions format that allows for more effective research assessment. Please use the free text boxes beneath each contributing author's name to add specific details on the author's contribution. More information is available in our guide to authors:

<https://www.embopress.org/page/journal/17574684/authorguide#authorshipguidelines>

- Indicate in legends exact n and exact p values, not a range, along with the statistical test used. To keep the figures "clear" some authors found providing an Appendix table Sx with all exact p-values preferable. You are welcome to do this if you want to.

- In Methods, provide the antibody dilutions that were used for each antibody.

- Please include structured Methods section that includes a Reagents and Tools Table (should be uploaded as a separate file) followed by a Methods and Protocols section. More information on how to adhere to this format as well as downloadable templates (.docx) for the Reagents and Tools Table can be found in our author guidelines:

<https://www.embopress.org/page/journal/17574684/authorguide#structuredmethods>

An example of a paper with Structured Methods can be found here:

<https://www.embopress.org/doi/full/10.1038/s44320-024-00037-6#sec-4>

- Please add data availability statement. If no data were deposited in public repositories include the sentence "This study includes no data deposited in external repositories."

4) Appendix: Please rename "Supplementary protocols" to "Appendix" with the table of content on the title page. Update the reference in the main manuscript text. Remove "Supplementary Information - Chemical Synthesis of C26 and analogues" and "Results and Discussion" from the title page and rename "Supplementary Scheme 1" etc. to "Appendix Figure S1" etc. Please update callouts in the text.

5) Expanded View Figures: Please add the expanded view figure legends to the main manuscript text, after the main figure legends and correct the nomenclature to "Figure EV1" etc.

6) Funding: Please indicate sources of funding in our submission system and in "Acknowledgments" in the manuscript file.

7) The Paper Explained: Please provide "The Paper Explained" and add it to the main manuscript text. Please check "Author Guidelines" for more information. <https://www.embopress.org/page/journal/17574684/authorguide#researcharticleguide>

8) Synopsis: Every published paper now includes a 'Synopsis' to further enhance discoverability. Synopses are displayed on the journal webpage and are freely accessible to all readers. They include separate synopsis image and synopsis text.

- Synopsis image: Please provide a visual abstract as a high-resolution jpeg file 550 px-wide x 300-600 pixels high to illustrate your article.

- Synopsis text: Please provide a short standfirst (maximum of 300 characters, including space) as well as 2-5 one sentence bullet points that summarise the paper as a .doc file. Please write the bullet points to summarise the key NEW findings. They should be designed to be complementary to the abstract - i.e. not repeat the same text. We encourage inclusion of key acronyms and quantitative information (maximum of 30 words / bullet point). Please use the passive voice.

9) Source data: Upload source data as one zipped folder per figure. Please note that we do not allow powerpoint as a file format for figures or for source data files, all .ppt file should be converted to PDF. Also, make sure that source data are complete. Currently source data for Figures 1B, 3B, 4A are missing.

10) As part of the EMBO Publications transparent editorial process initiative (see our Editorial at <http://embomolmed.embopress.org/content/2/9/329>), EMBO Molecular Medicine will publish online a Review Process File (RPF) to accompany accepted manuscripts. This file will be published in conjunction with your paper and will include the anonymous referee reports, your point-by-point response and all pertinent correspondence relating to the manuscript. Let us know whether you agree with the publication of the RPF and as here, if you want to remove or not any figures from it prior to publication. Please note that the Authors checklist will be published at the end of the RPF.

11) Please provide a point-by-point letter INCLUDING my comments as well as the reviewer's reports and your detailed responses (as Word file).

I look forward to reading a new revised version of your manuscript as soon as possible.

Yours sincerely,

Zeljko Durdevic

Zeljko Durdevic
Senior Editor
EMBO Molecular Medicine

*** Instructions to submit your revised manuscript ***

1) a .docx formatted version of the manuscript text (including Figure legends and tables)

2) Separate figure files*

3) supplemental information as Expanded View and/or Appendix. Please carefully check the authors guidelines for formatting Expanded view and Appendix figures and tables at <https://www.embopress.org/page/journal/17574684/authorguide#expandedview>

4) a letter INCLUDING the reviewer's reports and your detailed responses to their comments (as Word file).

5) The paper explained: EMBO Molecular Medicine articles are accompanied by a summary of the articles to emphasize the major findings in the paper and their medical implications for the non-specialist reader. Please provide a draft summary of your article highlighting

6) Author contributions: the contribution of every author must be detailed in a separate section.

7) EMBO Molecular Medicine now requires a complete author checklist (<https://www.embopress.org/page/journal/17574684/authorguide>) to be submitted with all revised manuscripts. Please use the checklist as guideline for the sort of information we need WITHIN the manuscript. The checklist should only be filled with page numbers where the information can be found. This is particularly important for animal reporting, antibody dilutions (missing) and exact values and n that should be indicated instead of a range.

8) Every published paper now includes a 'Synopsis' to further enhance discoverability. Synopses are displayed on the journal webpage and are freely accessible to all readers. They include a short stand first (maximum of 300 characters, including space) as well as 2-5 one sentence bullet points that summarise the paper. Please write the bullet points to summarise the key NEW findings. They should be designed to be complementary to the abstract - i.e. not repeat the same text. We encourage inclusion of key acronyms and quantitative information (maximum of 30 words / bullet point). Please use the passive voice. Please attach these in a separate file or send them by email, we will incorporate them accordingly.

You are also welcome to suggest a striking image or visual abstract to illustrate your article. If you do please provide a jpeg file 550 px-wide x 300-600px high.

9) A Conflict of Interest statement should be provided in the main text

10) Please note that we now mandate that all corresponding authors list an ORCID digital identifier. This takes <90 seconds to complete. We encourage all authors to supply an ORCID identifier, which will be linked to their name for unambiguous name identification.

Currently, our records indicate that the ORCID for your account is 0000-0001-5178-1175.

Link Not Available

11) Include a Reagents and Tools Table as part of the Methods section, which can be downloaded from our author guidelines (<https://www.embopress.org/page/journal/17574684/authorguide#structuredmethods>)

Photos 400-800 DPI

*Additional important information regarding figures and illustrations can be found at <https://bit.ly/EMBOPressFigurePreparationGuideline>. See also figure legend preparation guidelines: <https://www.embopress.org/page/journal/17574684/authorguide#figureformat>

***** Reviewer's comments *****

Referee #1 (Remarks for Author):

The revised manuscript is suitable for publication.

The authors addressed the remaining editorial issues.

17th Jul 2025

Dear Dr. Puthalakath,

We are pleased to inform you that your manuscript is accepted for publication and is now being sent to our publisher to be included in the next available issue of EMBO Molecular Medicine.

Zeljko Durdevic
Senior Editor
EMBO Molecular Medicine
